# 3D Printing of Thermoplastic Elastomers: Role of the Chemical Composition and Printing Parameters in the Production of Parts with Controlled Energy Absorption and Damping Capacity

**DOI:** 10.3390/polym13203551

**Published:** 2021-10-15

**Authors:** Marina León-Calero, Sara Catherine Reyburn Valés, Ángel Marcos-Fernández, Juan Rodríguez-Hernandez

**Affiliations:** 1Adática Engineering, Av. Leonardo Da Vinci, 8, Oficina 216, 28906 Getafe, Spain; marina.leon@adatica.com; 2Institute of Polymer Science and Technology (ICTP-CSIC), Juan de la Cierva, 3, 28006 Madrid, Spain; saracrv123@gmail.com; 3Interdisciplinary Platform for “Sustainable Plastics towards a Circular Economy” (SUSPLAST-CSIC), 28006 Madrid, Spain

**Keywords:** thermoplastic elastomers, fused deposition modeling, polymer characterization, mechanical properties, energy absorption, damping capacity

## Abstract

Additive manufacturing (AM) is a disruptive technology that enables one to manufacture complex structures reducing both time and manufacturing cost. Among the materials commonly used for AM, thermoplastic elastomers (TPE) are of high interest due to their energy absorption capacity, energy efficiency, cushion factor or damping capacity. Previous investigations have exclusively focused on the optimization of the printing parameters of commercial TPE filaments and the structures to analyse the mechanical properties of the 3D printed parts. In the present paper, the chemical, thermal and mechanical properties for a wide range of commercial thermoplastic polyurethanes (TPU) filaments were investigated. For this purpose, TGA, DSC, ^1^H-NMR and filament tensile strength experiments were carried out in order to determine the materials characteristics. In addition, compression tests have been carried out to tailor the mechanical properties depending on the 3D printing parameters such as: infill density (10, 20, 50, 80 and 100%) and infill pattern (gyroid, honeycomb and grid). The compression tests were also employed to calculate the specific energy absorption (SEA) and specific damping capacity (SDC) of the materials in order to establish the role of the chemical composition and the geometrical characteristics (infill density and type of infill pattern) on the final properties of the printed part. As a result, optimal SEA and SDC performances were obtained for a honeycomb pattern at a 50% of infill density.

## 1. Introduction

Additive manufacturing (AM), in contrast to conventional technologies such as injection or extrusion, involves layer-by-layer construction, so that upon the subsequent addition of multiple layers a 3D structure is obtained. In comparison to the abovementioned techniques, AM enables the fabrication of highly complex structures defined by a CAD design, in a shorter period of time and, with a clear material waste reduction. AM generally is classified in eight main categories [1] where fused deposition modelling (FDM), which consists of material extrusion, is the most popular because of its cost-effectiveness, wide range of material availability, and its capability to manufacture parts having good mechanical properties [2]. Time reduction and lower manufacturing costs are other additional advantages provided by this technology.

Among the materials used for AM, in addition to metals and ceramics, polymers have become a centre of interest for a wide range of applications. The versatility and synthetic adaptability, as well as the wide range of properties that can be achieved with polymers, have made of them the most used materials for AM methodologies. However, research using polymers have been mostly limited to the use of certain commodity and high performance thermoplastics [3]. Acrylonitrile butadiene styrene (ABS) and polylactic acid-based (PLA) thermoplastics have been the primary filament materials since the beginning of the FDM technology [4]. As a result, most studies in the literature have focused on those materials evaluating the mechanical behaviour of FDM-produced parts with respect to the printing parameters (such as layer height, infill %, part on bed orientation, bed and nozzle temperature or pattern among others) [5,6,7,8,9].

More recently, the use of thermoplastic elastomers (TPE) with FDM, which enables the fabrication of 3D printed parts with new properties, has also been investigated. Thermoplastic elastomers are of high interest due to their ability to stretch to moderate elongations and return to its near original shape, energy absorption capacity [10,11,12,13], energy efficiency [10,11], cushion factor [14,15] or damping capacity [14]. It is worth mentioning that thermoplastic elastomer parts can be also manufactured by Selective Laser Sintering (SLS) technology. However, SLS presents some disadvantages with respect to FDM such as higher cost, low recyclability, more complex processability or design limitations (such as in the case of the design of parts that require closed cavities) [16]. In contrast to PLA and ABS, few investigations have focused on the evaluation of 3D printed parts using thermoplastic elastomers [4,17,18,19,20]. However, several examples of applications can be already found in the literature for soft pneumatic actuators [21,22,23], biomedical applications [24,25,26] or flexible grippers [27,28] among others that permit to foreseen a wide range of uses for 3D printed parts using TPE. 

Thermoplastic elastomers are classified by their chemistry into six categories: thermoplastic polyurethanes (TPU), styrenic block copolymers (TPS), polyolefinic rubber blends (TPO), thermoplastic vulcanizates (TPV), thermoplastic polyetherimide (TPE) and thermoplastic polyether ester (TPC). Each category presents different chemical compositions and therefore produces parts with different properties. In spite of this variety of chemical alternatives, the fabrication of elastic parts from TPE has been mainly carried out using TPUs for applications such as biomedical products [29,30,31,32], sport equipment [33] or engineering parts [21,34,35]. More importantly, most of the studies employ commercial TPU materials without a clear understanding of the chemical structure of the material and therefore, the material contribution in the final properties of the printed part, i.e., the role of the materials chemistry has been until now neglected.

In this context, the present paper focused first on the analysis of the material chemical composition (type of monomers), the evaluation of the hard-segment content and the thermal properties of different commercial TPUs. Secondly, on the optimization of the printing experimental parameters to fabricate specimens with variable geometrical aspects (infill pattern type and infill density). Finally, this know how helped to understand the mechanical behaviour, in particular to produce parts with energy absorption and damping capacity. 

In this sense, it is worth mentioning that the demand is increasing for lightweight structures with high energy absorption and damping capacity for different applications in engineering fields such as transportation, aerospace, and civil engineering. However, until now most studies devoted to energy absorption focused on the optimization of the structural aspects. As a consequence, a number of energy absorbers with different structures such as columnar, sandwich, plates, honeycomb, and foams have been proposed in recent years. Although significant energy absorption capacity was achieved in these studies, the structures still need further optimization [36]. Moreover, the role of the material on the mechanical properties of the fabricated parts has been somehow ignored but, as will be described, may provide new opportunities to tailor and even improve the mechanical behaviour. Therefore, herein, the chemical characteristics of the material employed will be correlated with the mechanical properties of the final printed part.

## 2. Materials and Methods

### 2.1. Materials

There are several TPEs commercialized as filament for 3D printing. As it will be described later, three different chemical structures have been identified for these materials. In the following, the elastomeric commercial filaments employed in this study are listed:TPU-Ultimaker (Ultimaker, Utrecht, Netherlands) [37]Flexfil_93A (Filamentum, Hulín, Czech Republic) [38]FlexSmart (SmartMaterials, Jaén, Spain) [39]PolyFlex_95A (Polymaker, Shangai, China) [40]eFLEX (eSUN, Shenzhen, China) [41]Innovatefil TPU Hardness + (83D) Smart Materials 3D (Innovatefil, Jaén, Spain) [42]Filaflex_95A (Recreus, Alicante, Spain) [43]Filaflex_82A (Recreus, Alicante, Spain) [44]Filaflex_70A (Recreus, Alicante, Spain) [45]FlexiSmart (FFF World, Cantabria, Spain) [46]

### 2.2. FDM 3D Printers

Three different 3d printers were used depending on the Shore A hardness of the filament. Raise3D Pro2 and Pro2 Plus were used to print filaments of 95 A and 82 A hardness since both printers have the same print head system. For lower shore grades (70 A), Raise3D E2 was the best choice due to the shorter print head system. This is due to the fact that, Raise series Pro2 presents a longer duct from the feeding tube up to the extrusion head, and soft filaments do not have enough strength to compensate the pressure generated in the nozzle. The narrowing of the duct results in clogging of the filament causing 95blockage.

### 2.3. Chemical Composition of the Filaments–^1^H-NMR

The chemical structure and the quantitative composition of the TPUs were determined by proton nuclear magnetic resonance (^1^H-NMR). Approximately 10 mg of the samples were dissolved in 1 mL of deuterated DMSO (DMSO-d6). Heating was necessary to dissolve the samples, and in the case of white filaments (TPU-Ultimaker, FlexiSmart, PolyFlex_95A, Filaflex_95A) the solution was previously filtered to remove the inorganic white pigment. Spectra were acquired on a Unity Plus 400 instrument (Varian Inc, Palo Alto, CA, USA) at room temperature. All recorded spectra were referenced to the residual solvent signal at 2.50 ppm.

### 2.4. Dynamic Scanning Calorimetry (DSC) 

The thermal transitions of the samples were analysed by differential scanning calorimetry (DSC) on a DSC 822e calorimeter (Mettler Toledo, Schwerzenbach, Switzerland) equipped with a liquid nitrogen accessory. Filament samples were pressed at 150 °C and discs cut from the pressed samples weighing approximately 10 mg were sealed in aluminium pans with perforated lid. Samples were heated, from −80 to 200 °C at a rate of 10 °C·min^−^^1^, cooled to −80 °C at the maximum rate of the instrument, maintained for 5 min at this temperature and re-heated from −80 to 200 °C at a rate of 10 °C·min^−^^1^.

Melting temperatures (Tm) and crystallization temperatures (Tc) are given as the maximum of the endothermic transition and the minimum of the exothermic transition, respectively. Glass transition temperature (Tg) was calculated as the midpoint of the change in heat capacity. In addition, the start and the end of the Tg were registered. 

### 2.5. Thermogravimetric Analysis (TGA)

The thermogravimetric analysis (TGA) of samples was carried out in a Mettler-Toledo TGA/SDTA 851 instrument from room temperature to 600 °C under a nitrogen atmosphere at a 10 °C·min^−1^ heating rate. Filament samples were pressed at 150 °C and discs cut from the pressed samples weighing approximately 10 to 18 mg.

### 2.6. Filament Mechanical Properties

The mechanical properties in tension of the filaments were measured in a MTS Synergie 200 testing machine (MTS Systems Corporation, Eden Prairie, MN, USA) equipped with a 100 N load cell. All the test specimens analysed were cut directly from the commercial filaments with a length of 40 mm. A cross-head speed of 200 mm·min^−1^ was used and the strain was measured from cross-head separation and referred to a 10 mm initial length. A minimum of five specimens were analysed.

### 2.7. Compression Tests

Compression strength was analysed following the ISO 7743 standard. Five cylinders of 29 ± 0.5 mm diameter and of 12.5 ± 0.5 mm height were tested for each sample. In addition, regarding the infill percentage, 3 top and bottom layers were manufactured in all test pieces to ensure planar contact with metal plates. The specimens were tested at a speed of 10 mm·min^−^^1^ until a strain of 25% was reached. Then, the strain was released at the same speed and this procedure was repeated three more times, so four compression cycles were performed in a continuous sequence. Depending on the infill percentage, the compression force varied very widely, so the load cell was varied accordingly (100, 1000 and 5000 N). These mechanical tests allow to calculate mechanical properties such as the compression modulus at different elongation conditions, hysteresis, energy absorption capacity or specific damping capacity for each material, pattern and infill densities.

### 2.8. 3D Printing of the Compression Specimens 

Fused deposition modelling (FDM) is a 3D printing process in which a thermoplastic or a thermoplastic elastomer (in our case, a thermoplastic polyurethane) passes through a heating element that partially melts the material. This semi-molten material is then dropped through a nozzle which can move in the XY plane, onto a platform. The deposition takes place according to a path which is generated by the slicing software. After completing one layer of deposition, the part moves vertically in the Z plane and the process is repeated until the part is finished [47].

The compression specimens were fabricated using the materials listed above. Extrusion temperature, printing speed and heat bed temperature are the main parameters that were optimized to improve the quality of the printed parts. In Table 4, the most relevant parameters employed in this study are summarized.

In addition to the printing parameters, different geometrical parameters were investigated to produce parts with modulated mechanical properties. Together with the different properties of each material, the type of infill and the infill density were systematically varied. 

With the appropriate .stl file in hand, the IdeaMaker software was used as slicer to generate the gcodes. IdeaMaker slicer provides the possibility to control the infill density and select the type of pattern of the infill. In this study, grid, honeycomb and gyroid patterns were chosen. Grid and honeycomb infill structures are among the most extensively employed to create non-solid 3D printed parts. For instance, honeycomb is a structure widely employed, among others, in aircraft structures due to their good mechanical response and lightweight [48,49]. The gyroid pattern, which was discovered by NASA [50], is lightweight and presents high-strength properties [51,52]. It is interesting to note that, all these internal structures are rather difficult if not impossible to obtain using conventional manufacturing technologies. 

In order to modulate the mechanical properties of the specimens, in addition to the use of different materials (different chemical composition) and variable hardness (ranging from 70 A to 95 A), the parts were fabricated with different infill density percentages: 10, 20, 50 and 80%. In addition, for comparative purposes, solid specimens (100% infill) were fabricated from each material. In this case, a grid pattern was selected to generate the structure. As it will be described, for a single material both the structure and infill density will enable to finely tune the mechanical response of the 3D printed part.

## 3. Results and Discussion

### 3.1. Chemical Characterization of the Thermoplastic Elastomers

Several thermoplastic elastomers (TPE), with variable composition and hardness, were employed to study the fabrication of 3D printed parts with variable energy absorption and damping properties. While the materials selected are all thermoplastic polyurethanes (TPU), in order to understand the material properties, it is critical to carry out a chemical characterization of the thermoplastic elastomers. ^1^H-NMR was employed for this purpose since this technique provides precise information about the monomer chemical composition. ^1^H-NMR served to identify the TPU components, i.e., the type of difunctional macroglycol employed, the diisocyanate and the difunctional chain extender. Moreover, ^1^H-NMR allowed quantification by analysing the value of the proton integrals, to determine the molar ratio of the monomers in each TPU. This molar ratio is, in turn, directly related to the mechanical properties of the material.^1^H-NMR demonstrated that, in all cases, the filaments were prepared by the reaction of a macroglycol of different chemical structure, 4,4’-methylene-bis(phenyl diisocyanate) (MDI) as diisocyanate and butanediol (BD) as chain extender. Three different macroglycols were identified: poly(butylene adipate) diol (PBA), poly(ethylene adipate) diol (PEA) and poly(tetramethylene oxide) diol (PTMG). In Figure 1 and Figure 2 illustrative ^1^H-NMR examples of the TPUs composed of different macroglycols are presented. The peak at 9.49 ppm (slightly shifted to 9.63 ppm for Flexi Smart) corresponds to NH protons of urethane groups. The aromatic peaks at 7.35, 7.33, 7.08 and 7.06 ppm and the aliphatic peak at 3.80 ppm correspond to the protons from the reacted 4,4′-methylene-bis(phenylisocyanate) (MDI). Peaks at 4.09 and 1.69 ppm correspond to 1,4-butanediol reacted with isocyanate groups (BD extender) and therefore with urethane groups at both ends. All these peaks are common for all TPU samples. However, some differences can be identified depending on the macroglycol employed. For the PBA and PEA based TPUs, peaks at 2.28 and 1.51 ppm related to adipate molecules (Adipate) were found. In the case of PTMG based TPUs two peaks at 3.30 (submerged in the water signal) and at 1.48 ppm related to tetramethylene glycol (PTMG) molecule were identified. In PBA based TPUs, peaks at 4.01, 1.65 and 1.59 ppm correspond to 1,4-butane diol reacted with adipate molecules (BD adipate) and therefore with ester groups at both ends, and in PEA based TPU, peaks at 4.24 and 4.20 ppm correspond to ethylene glycol (EG) reacted with isocyanate groups and with adipate molecules respectively. Signals at 3.31 and 2.50 ppm are related to water (H_2_O) and residual signal of deuterated solvent (DMSO) respectively.

In summary, it is concluded that all the thermoplastic elastomers are TPU and that three different types of TPUs can be identified. Ultimaker, Flexfill, Smartfil and Polyflex were prepared using PBA as macroglycol. The materials eFLEX, Innovatefil TPU Hardness+, Filaflex (95 A), Filaflex (82 A) and Filaflex (70 A) were prepared using PTMG. Finally, Flexi Smart (88 A) involved the use of PEA as macroglycol. From these ^1^H-NMR results, a reaction scheme for all the different TPUs investigated is proposed (Figure 3).

In addition to the chemical composition, another important parameter in TPEs is the hard-segment content. TPUs are segmented polymers where the macroglycol chains constitute the so-called soft segments (SS) whereas the segments produced by the reaction of the diisocyanate MDI with the chain extender BD constitute the so-called hard segments (HS), as seen in Figure 3. 

From the molar ratio, the hard-segment (HS) weight percent of the TPU, defined as:(1)$$\%\mathrm{HS}=\frac{\mathrm{weight}\;\mathrm{MDI}+\mathrm{weight}\;\mathrm{BD}}{\mathrm{total}\;\mathrm{weight}}\times 100$$
could be determined. The HS content (along with the length of the macroglycol, that could not be determined from ^1^H-NMR spectra) strongly influences the mechanical properties of TPUs and is, for example, directly related with the hardness, i.e., higher hardness is observed by increasing the HS content. More interestingly, it is possible to modulate the hardness depending, among others, on the ratio of hard segments to soft segments. This is the case of the Filaflex that can be obtained with different hardness: 95 A, 82 A and 70 A. In Figure 2, the ^1^H-NMR spectra of these three TPUs are presented. Using the PTMG signals as reference and set to the same intensity, it is clear that urethane, MDI and BD chain extender peaks increase with the increase in hardness.

In Table 1, the chemical composition as well as the %HS for all the commercial TPUs employed in this study are listed. Most of the filaments presented %HS between 42 and 54 with Shore A hardness ranging from 93–98 A. In addition, two particular cases require further consideration. On the one hand, Innovatefil TPU Hardness + has a %HS largely above the rest of the filaments, i.e., 85%. Therefore, this filament is a rather rigid material with a Shore of 83D, thus this material is flexible but presents a limited elastomeric character. On the other hand, as it has been already anticipated, Filaflex can be found with different %HS (between 31 and 54 %HS) and, as a consequence, the resulting materials present hardness ranging from 70 A to 95 A. Notice that, independently of the Shore hardness, the %HS has an important impact on the mechanical properties of the 3D printed parts providing, for example, higher strength for a higher % of the hard segment.

### 3.2. Thermal and Mechanical Properties of the Thermoplastic Elastomers 

The thermal behaviour of the different thermoplastic elastomers was investigated by dynamic scanning calorimetry (DSC) and thermogravimetry (TGA). A summary of the thermal properties of each filament is indicated in Table 2. On the one hand, DSC allowed us to determine the thermal characteristics of the material (Tg, crystallization and melting processes) which is particularly interesting to optimize the 3D printing fabrication conditions. In Figure 4, the DSC traces of illustrative examples of each type of TPU class (i.e., prepared using different macroglycols) and also, the Filaflex TPUs with variable hardness, are represented. The materials presented a Tg that varied depending on the type of material and the hardness. For instance, Innovatefil with a hardness of 83D showed a Tg of around +50 °C. The softest material, Filaflex 70 A, presented a Tg of around −60 °C. These differences allow to finely tune the range in which the material presents elastic properties by modulating the chemical composition. This can be also evidenced by focusing on materials (7), (8) and (9), i.e., Filaflex with variable hardness. In this case, the TPUs are constituted by the same chemical components but with different ratio and, as a result, the Tg gradually varied by from −37 °C down to −60 °C simply by decreasing the hard segment content in the TPU. 

Another important information obtained from DSC is related to the crystallization and fusion processes that establish the temperatures of use of these materials. Independently of the chemical composition and the TPU hardness all the DSC presented a rather broad fusion range (from 10–30 °C to 200 °C in most of the cases) of the hard segment with more than one maximum. 

On the other hand, TGA allowed us to determine the maximum temperature to be employed during the fabrication without material degradation. The TGA traces are represented in Figure 4 and clearly indicate that all TPUs were stable up to 240 °C. Above this range of temperatures thermal decomposition takes place in two distinct steps. As it has been described in literature, polyurethanes usually decompose in two main steps with the first step due to the decomposition of the urethane groups in the main chain and the second to the decomposition of the rest of the material (i.e., C-C and C-O bonds) [53]. While this occurs for all the TPUs some differences were observed depending on the chemical composition. More precisely, the TPUs formed using PBA as macroglycol presented the higher degradation temperatures in the range of 340–350 °C. Those prepared from PTMG presented Td1 in the range of 320–340 °C (for the range of hardness between 87 A and 95 A) and finally the TPU prepared from PEA presented the lower value at around 317 °C. It is also interesting to note that for the same chemical composition a reduction in the HS content has associated a decrease in the Td1. This can be clearly evidenced in the case of the different Filaflex. Filaflex 85 A has a Td1 of ~341 °C, which is reduced to ~327 °C for the case of Filaflex 82 A and finally to 298 °C for Filaflex 70 A.

In addition to the thermal properties, the mechanical properties of the materials in the form of filament (1.75 mm diameter) were investigated. Table 3 summarizes the mechanical properties in tension, i.e., modulus, tensile strength and elongation at break for the different TPU tested, and in Figure 5a illustrative strain-stress curves for selected examples of each TPU group (prepared using a different macroglycol) with similar hardness in the range of 88 A to 95 A are depicted. FlexiSmart presented a lower modulus but a larger elongation at break. Interestingly, PolyFlex (95 A) and FilaFlex (95 A) with different macroglycol structure and %HS, presented rather similar strain-stress curves. For the case of FilaFlex with variable hardness ranging from 95 A to 70 A (Figure 5b) a gradual decrease on both Modulus and tensile strength can be observed. Moreover, a decrease on the hardness has associated an increase on the elongation at break from around 610% up to 1200%. 

### 3.3. Optimization of the Fabrication Parameters for the TPUs

For the following sections, three TPUs chosen from each chemical type with similar hardness (Polyflex (95 A), Filaflex (95 A) and FlexiSmart (88 A)), and Filaflex with variable hardness from 95 A to 70 A, were selected to understand their mechanical behaviour in 3D printed parts. Prior to the mechanical characterization of 3D printed parts, an optimization of the manufacturing parameters was carried out. The printing temperature, the use of retraction or the printing speed are some of the critical parameters to be optimized to manufacture high quality flexible parts. 

In our particular case, the first parameter optimized was the printing temperature. In all cases, the maximal temperature to be employed was limited by the material degradation occurring above 240 °C as evidenced by TGA. Moreover, too low temperatures prevent material extrusion and produced clogging. For this reason, the temperatures selected (with slight variations for each material) were in the range of 225 to 235 °C. Another relevant parameter is the retraction. Retraction is usually employed in 3D printing of thermoplastics which reduces the formation of the so-called stringing or oozing between different areas of the same part or between different parts. However, when using elastic materials, the use of retraction should be strongly limited if not completely avoided. Retraction can be easily regulated in rigid thermoplastics but it is difficult to control in thermoplastic elastomers since TPE can suffer stretching during the extrusion process. In this sense, an alternative is the reduction of the printing speed in order to reduce stringing while allowing for a control of the material deposition without using retraction. Moreover, for soft materials, low extrusion speed is required to avoid blockage of the nozzle and also to improve the printing quality. To illustrate the effect of the printing speed on the quality, Figure 6 shows the comparison of gyroid pattern for 80% of infill density at different manufacturing speeds using Filaflex 82 A. For this particular material, at printing speeds of 10 and 15 mm·s^−1^ either non-homogeneous deposition (10 mm·s^−1^) or even defects in the structure (15 mm·s^−1^) can be observed (green arrows). However, further reduction of the printing speed down to 8 mm·s^−1^ significantly improved the quality of the deposition. As a result of the optimization process, the main fabrication parameters employed for each material are shown in Table 4.

### 3.4. Mechanical Properties of Compresssion Test Specimens: Role of the Type of Infill and Infill Density

Using the optimized printing experimental parameters, specimens for compression tests were fabricated by varying the type of infill and the infill density for the different materials selected. It is worth mentioning that 3D printing offers important advantages and some limitations, in comparison to traditional polymer processing approaches such as injection molding, to fabricate parts with modulated mechanical resistance. FDM based on filaments extrusion have presented important limitations when printing soft thermoplastic elastomers (such as TPUs), i.e., TPUs with hardness values below 60–70 A. However, as it will be described later, taking advantage of the versatility of the 3D printing, it is possible to reduce the compression modulus of the printed parts by changing the infill density and the type of infill. 

In order to understand the role of the infill density and the type of infill on the mechanical properties of the printed part for the materials selected, compression test specimens (i.e., cylinders of diameter 29 ± 0.5 mm and height of 12.5 ± 0.5 mm) were fabricated following the ISO 7743 standard. Figure 7 shows an IdeaMaker maker top view of the part to be printed and a photograph of the real printed parts after 10 deposition layers. As it can be observed three type of structures grid, gyroid and honeycomb were explored, and the infill density was varied between 10% and 80%. For comparative purposes, solid (100%) parts were fabricated using the grid structure.

Once the specimens were fabricated, compression tests were carried out for each infill density, pattern type and material selected. Four complete compression cycles were carried out up to 25% deformation of the cylinder height with a deformation rate of 10 mm·min^−1^. According to previous reports [11,15] after 3 cycles and, at least up to 8 compression cycles the behaviour is almost identical. Therefore, unless otherwise stated the curves represented and analysed correspond in all cases to the 4th cycle.

The first aspect explored was the role of the type of infill and the type of material in the compression behavior for a particular infill density. In Figure 8 and Figure 9, the 4th compression cycle for specimens fabricated using a 20% infill density with variable infill type, and the specimens fabricated with different infill percentage using the grid infill type, are respectively represented. For the sake of clarity, the curves for 10%, 50% and 80% infill density with other infill patterns are included separately in Appendix A. 

As showed in Figure 8 and Figure 9, clear inflection points were observed in the compression stress-strain curves for samples with 10% and 20% infill but not for those with higher infill densities 50, 80 and 100%. The compression curves of the grid pattern in both 10% and 20% presented a clear plateau whereas for gyroid and honeycomb with the same infill density the curves gradually grew. However, a change in the slope at around 8% of compression was clearly observed in the gyroid pattern and occurred more gradually in the samples manufactured using honeycomb pattern. Interestingly, the plateau observed was higher for Filaflex (0.3 MPa) than for Polyflex (0.24 MPa) and finally for FlexiSmart (0.16 MPa). 

While this behavior was expected for FlexiSmart (it has a lower Shore hardness) the differences observed in the curves for Filaflex and PolyFlex with similar hardness and Modulus were, a priori, unexpected. This behavior can be explained by the fact that both hardness and Modulus are properties related to small initial deformations. However, these differences in the plateau height can be related to the hard segment content which is significantly higher for Filaflex (54.2%) than for Polyflex (44.6%). Furthermore, FlexiSmart presented lower strength at compression than Polyflex which can be related to the lower %HS content of FlexiSmart (41.6%). Another interesting observation is related to the curves in the Filaflex series. By decreasing the hardness, as well as the %HS content, of the filament employed, the plateau height gradually decreased. For instance, in the case of the grid infill pattern, the plateau height decreased from 0.3 MPa using the 95 A filament, to around 0.1 MPa for the 80 A and finally to 0.06 MPa for the 70 A. In addition, independendly on the Shore hardness and %HS, the infill pattern type had an interesting impact on the strength results where the honeycomb highly increased the compression values.

In addition to the shape of the curves measured at low infill density, an increase of the infill percentage has also important effects in the slope of the curves (Figure 9) and the mechanical resistance of the printed part. Independently of the material employed and the type of pattern selected, an increase of the infill density produces stiffer parts in which the compression strength measured for a particular strain gradually increases with the infill. This is not surprising but it is worth mentioning that the control over the infill percentage allows to tailor the mechanical behaviour of the part. 

By analysing the Filaflex series, it can be concluded that similar compression strengths can be obtained from filaments with different hardness but applying a higher strain. For instance, using as example the GRID 50% to obtain parts able to achieve a compression strength of 0.5 MPa, this can be obtained using the same pattern, the same infill % and using either Filaflex 95A or Filaflex 82A but applying a 9% deformation or a 20% deformation.

Another important aspect is related to the weight of the fabricated parts. In this sense, in those cases where this aspect is critical, AM by FDM presents important advantages over traditional fabrication approaches. Instead of using softer materials, the same mechanical resistance can be obtained using more rigid materials but reducing the infill density.

From the compression tests depicted above, it was possible to calculate the compression modulus which measures the stiffness of the material or, in other words, the ability of the material to withstand changes in length when subjected to compressive loads. As depicted in Figure 10 and Figure 11, two values of compression modulus were calculated for 10% and 20% deformation respectively by using the equation in Figure 10. These moduli have been calculated for the different infill patterns, infill densities and materials.

The first interesting conclusion obtained from these graphs is that the compression modulus at 10% strain was higher than at 20% for infill densities of 10% and 20%. This can be explained by taking into account that in this range of infills a plateau region is observed. Higher densities resulted in all cases in higher modulus at 20% than at 10%. This behaviour changed for infill densities of 50% and above where the values of compression moduli at 10% were in all cases below the values of those measured for infills of 20%.

The second relevant aspect is that, an increase of the infill density (independently of the type of pattern) produced and increase of the modulus measured both at 10 and 20% of strain.

Also, the type of infill pattern appears to be more relevant in specimens with lower infill density. For instance, for infill densities of 10% and 20% grid and honeycomb patterns produced specimens with similar compression modulus, while the compression modulus obtained for specimens with gyroid pattern were clearly below. At least to some extent, this can be explained by the fact that gyroid structure is easier to be compressed. it can be related to the vertical wall that oppose to the compression direction for grid and honeycomb. Interestingly, these differences disappeared for higher infill densities (50–80%).

Finally, by focusing in Figure 11, as expected, the use of materials with different hardness in combination with the control over the infill percentage allowed for a fine control over the compression modulus of the printed part. In effect, by considering these two characteristics, specimens with precisely targeted compression modulus can be designed and fabricated. It is interesting to note that, although Filaflex95A and Polyflex95A have the same Shore hardness, the compression modulus showed slight differences which might be associated to the higher %HS of Filaflex95A (54.2%) respect to Polyflex95A (44.6%)

### 3.5. Energy Absorption of the TPU 3D Printed Parts

With the knowledge of the stress and strain, the energy absorption capacity can be studied. In particular, this property provides information about the capability of a part to absorb the force applied over a surface without the structure collapse. It is worth mentioning that the results of energy absorption can be used to guide the structural design and the optimization for a high capacity of energy absorption [54]. More precisely, the absorbed energy per unit volume in the loading process is the area under the stress–strain curve (see Figure 12). Furthermore, in order to normalize the absorption capacity independently on the material density, specific energy absorption (SEA) has been calculated following the equation:$$\mathrm{SEA}\left(J\right)=W_{absorbed}=\int_{0}^{\varepsilon }\sigma \left(\varepsilon \right)d\varepsilon$$

Firstly, SEA capacity along each cycle (up to the fourth cycle) has been calculated (see Figure A5 in Appendix A). A decrease in the absorption capacity for all the cases that is stabilized above the 3rd-4th cycle can be observed. However, it is interesting to note that for TPUs with lower hardness, the differences in energy absorption capacity between 1st and 4th cycle are significantly lower which means that the material permanent deformation (or compression set) due to the repetitive stress deformation applied is lower.

By using the curves of the 4th cycle, it is possible to plot the SEA as a function of the infill percentage (Figure 12) for all the materials explored. According to Figure 12A, Filaflex95A presents the highest absorption capacity that gradually decreases for PolyFlex 95 A and finally for FlexiSmart 88 A. In the Filaflex series (Figure 12B) a similar trend is observed with a significant decrease of the SEA by decreasing the material hardness, in particular, from 95 A to 82 A–70 A.

Another relevant information obtained from Figure 12 is that, by increasing the infill percentage, an increase of the SEA is observed. However, this increase significantly depends on the type of material (and its related %HS content) and the infill pattern. For instance, by analysing Filaflex 95 A, the specimens fabricated using grid and gyroid patterns presented a gradual increase of the SEA as a function of the infill density. In the case of specimens fabricated with honeycomb infill patterns, the increase occurs faster by increasing the infill percentage up to 50% but the increase is limited above this percentage. A similar trend is observed for PolyFlex 95 A and FlexiSmart 88 A. 

Finally, it is possible to conclude that the honeycomb geometry provides, in most of the cases, the best SEA performance for all material and infill densities. The behaviour of this type of structure is particularly interesting in the range of 50% of infill density. Other previous studies have also evidenced the excellent mechanical properties and various topological configurations of this structure [13].

### 3.6. Specific Damping Capacity

The specific damping capacity (SDC) of a structure is a constant value that provides the recovery capacity to the initial state. The SDC can be calculated from the ratio of dissipated energy (load-unload) to stored energy (load) following the equation shown in Figure 13. Hence, a higher SDC implies higher load-unload difference, so higher hysteresis and higher damping capacity [55]. In Figure 13, the SDC values are represented as a function of the infill density for the different materials selected and the three types of infill. 

By observing the general trends of the curves, it is possible to conclude that, in general, the samples prepared with a 10% infill density have lower SDC but upon increasing above 20% of infill the values of SDC tend to stabilize. Thus, in the range of 20% and up to 80 and even 100% of infill density, SDC remains in a similar range.

Another interesting aspect is related to the type of infill. Following a similar trend in comparison to SEA, the honeycomb structures produced the higher SDC values which imply better damping performance. Moreover, the best values were obtained for honeycomb in the range of 20–80% of infill. Furthermore, it seems that an optimum point can be achieved for an infill density lower than 80%, which implies that SEA and SDC can be optimized while a weight reduction is achieved. 

Finally, also the material appears to play a key role. In the series of materials with similar Shore hardness (88 A–95 A), Filaflex 95 A presented the highest SDC while PolyFlex and, in particular, FlexiSmart SDC values were clearly below. This is a relevant result since, in addition to the Shore hardness, the chemical structure of the material together with the hard segment content appears to be determinant in the final properties of the printed part. The results obtained for the Filaflex series with different Shore hardness indicate that the SDC significantly decreased by using softer materials, so a higher soft segment content reduces structure damping capacity. 

## 4. Conclusions

A thorough chemical characterization of commercially available TPU materials employed for the fabrication of 3D printed elastic parts was carried out. Upon ^1^H-NMR analysis, three different types of TPUs were identified depending on the macroglycol employed. More precisely, Ultimaker, Flexfill, Smartfil and Polyflex were based on PBA macroglycol, the materials eFLEX, Innovatefil TPU Hardness +, Filaflex (95 A), Filaflex (82 A) and Filaflex (70 A) on PTMG macroglycol and finally Flexi Smart (88 A) on PEA macroglycol. This technique allowed also to quantify the hard segment content, composed in all cases of MDI and BD, which plays also a critical role in the final properties of the printed part.

DSC traces allowed the determination of both the temperature required to melt the material and the range in which the material remains in an elastic state (temperature range between the Tg and the Tm). Depending on the material, the Tg can decrease down to −60 °C for the softer materials. The melting process occurs during a wide range of temperatures indicating a multistep process. Equally, as evidenced by TGA, the degradation of the TPUs starts at around 240 °C which determines the maximal temperature to be employed during the printing process. 

Both, type of infill and infill density, influenced the compression behaviour of the 3D printed samples. Compression strength could be increased by increasing the infill density for all the patterns; and at all infill densities, compression strength was lower for gyroid pattern and higher for honeycomb pattern with intermediate values for grid pattern. For TPUs with the same chemical structure, compression strength increased with hardness increment, and for TPUs with the same hardness, compression strength increased with the increase on hard segment content. 

According to our findings three aspects resulted crucial to obtain the best SEA and SDC values, i.e., type of material (and thus the %HS content), the type of infill and the % of infill employed. Regarding the type of material, and specially its %HS content, it was observed that for higher %HS content (Filaflex95A), the energy absorption and damping capacity increased. In addition, for higher infill density, the absorption energy increased; however, the damping capacity improvement depended on the infill pattern and did not show a direct relation to the infill density. Lastly, honeycomb pattern showed the best results for SEA and SDC in the range of 20–80% of infill density.

In particular, Filaflex 95A combined with an infill density between 20–50% and using honeycomb as infill pattern provided parts with the optimal SEA and SDC properties. 

## Figures and Tables

**Figure 1 polymers-13-03551-f001:**
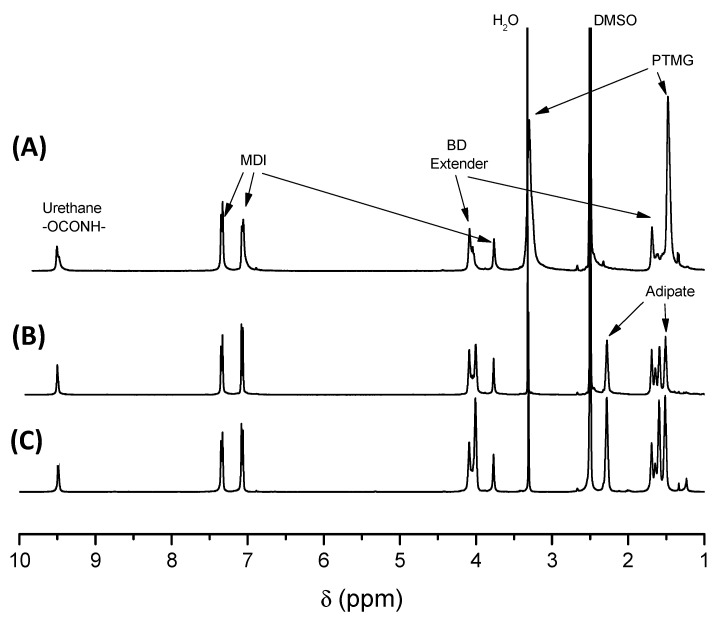
Illustrative ^1^H-NMR of a (**A**) TPU based on PBA+MDI+BD (Flexfil_93A), (**B**) TPU based on PTMG+MDI+BD (Filaflex_95A) and (**C**) TPU based on PEA+MDI+BD (FlexiSmart).

**Figure 2 polymers-13-03551-f002:**
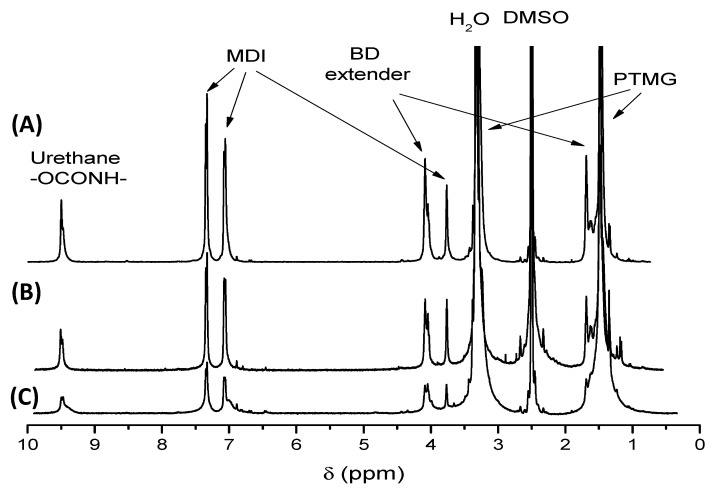
^1^H-NMR spectra of Filaflex samples with different hardness: (**A**) 95 A, (**B**) 82 A and (**C**) 70 A.

**Figure 3 polymers-13-03551-f003:**
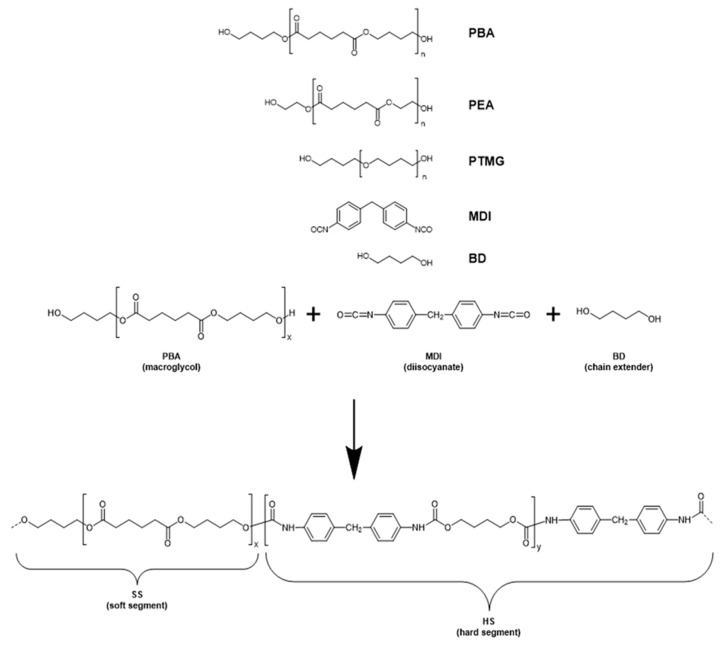
Reaction scheme for the preparation of the thermoplastic polyurethanes (TPU). Depending on the TPU three different macroglycols were employed, i.e., PBA, PEA and PTMG (structure depicted above). Below: scheme of the reaction involving PBA as macroglycol, MDI as diisocyanate and BD as chain extender.

**Figure 4 polymers-13-03551-f004:**
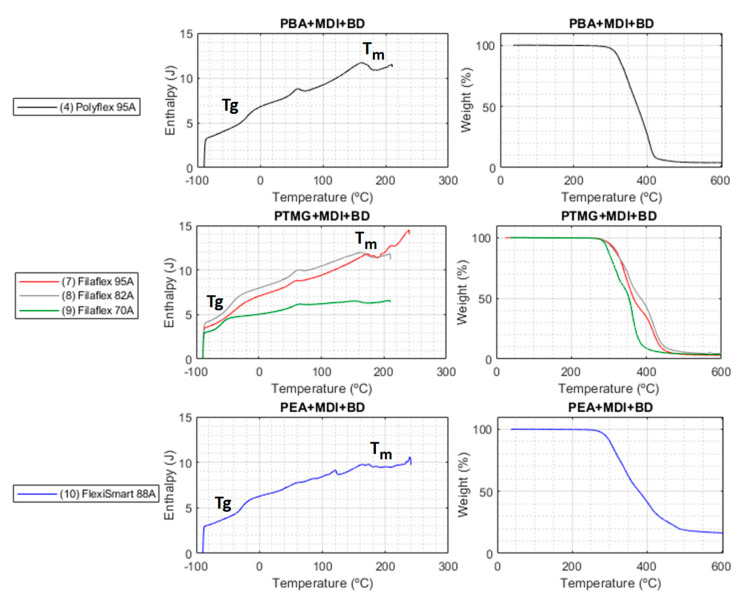
DSC (**left**) and TGA (**right**) plots for each macroglycol-based family of TPUs.

**Figure 5 polymers-13-03551-f005:**
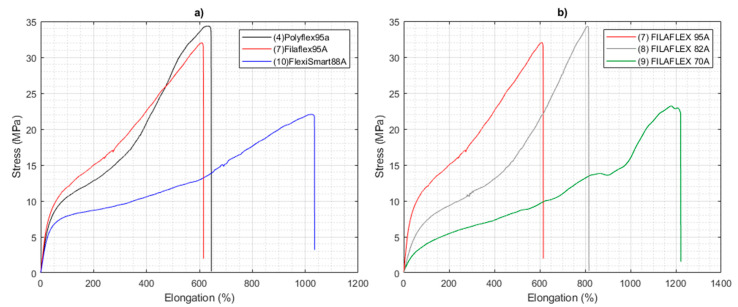
Stress-strain tests of the TPU filaments (**a**) Comparison of the different TPU classes as a function of the type of macroglycol employed, i.e., PolyFlex95A, Filaflex95A and FlexiSmart88A, (**b**) Comparison of the strain-stress curves for the same TPU class with variable hardness: 95 A, 82 A and 70 A.

**Figure 6 polymers-13-03551-f006:**
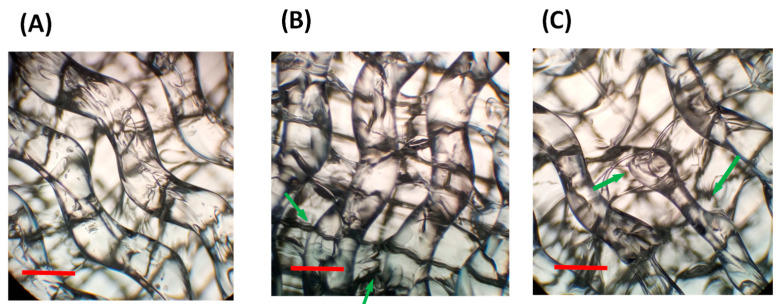
Optical microscope images of a gyroid pattern (infill density of 80%) fabricated using Filaflex 82A at a variable printing speed: (**A**) 8, (**B**) 10 and (**C**) 15 mm·s^−1^.

**Figure 7 polymers-13-03551-f007:**
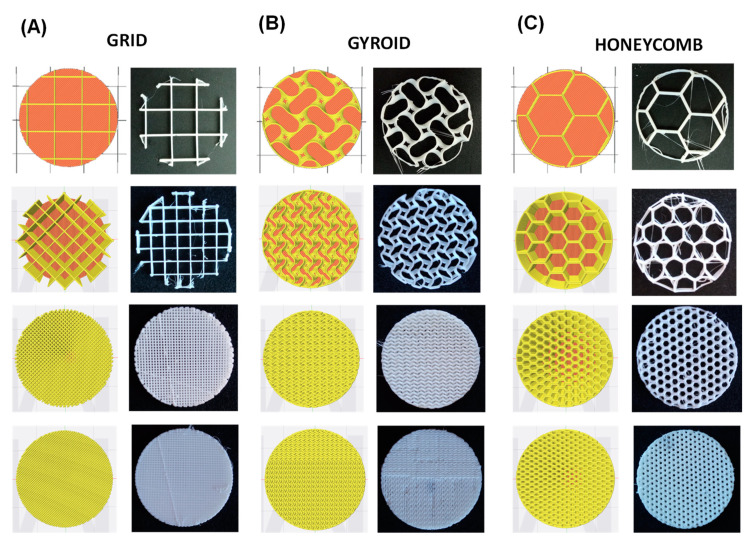
Illustrative designs and optical images of the types of infill and infill density of the patterns analyzed: (**A**) grid 10, 20, 50 and 80%, (**B**) gyroid 10, 20, 50 and 80%, and (**C**) honeycomb 10, 20, 50 and 80%.

**Figure 8 polymers-13-03551-f008:**
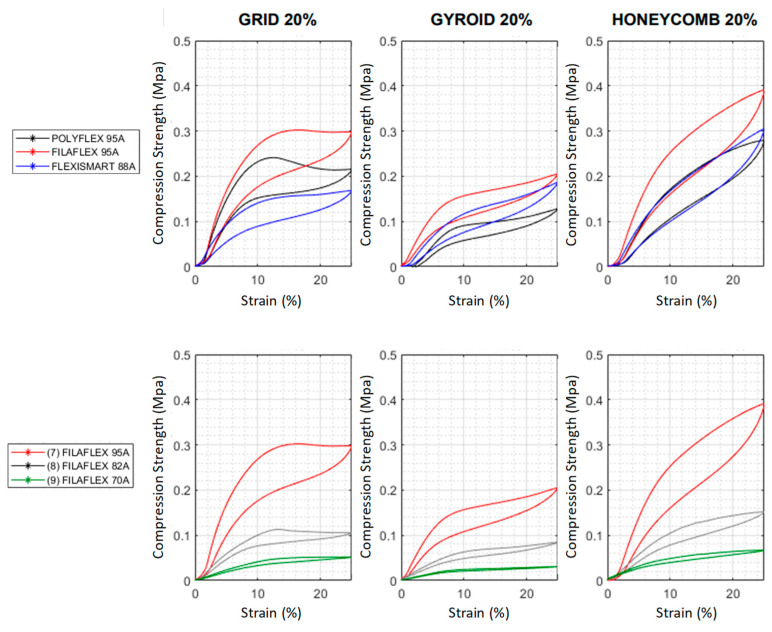
Compression stress curves obtained for the 4th cycle for different TPU materials. The samples were prepared with a 20% infill density with three different infill patterns (GRID, GYROID and HONEYCOMB).

**Figure 9 polymers-13-03551-f009:**
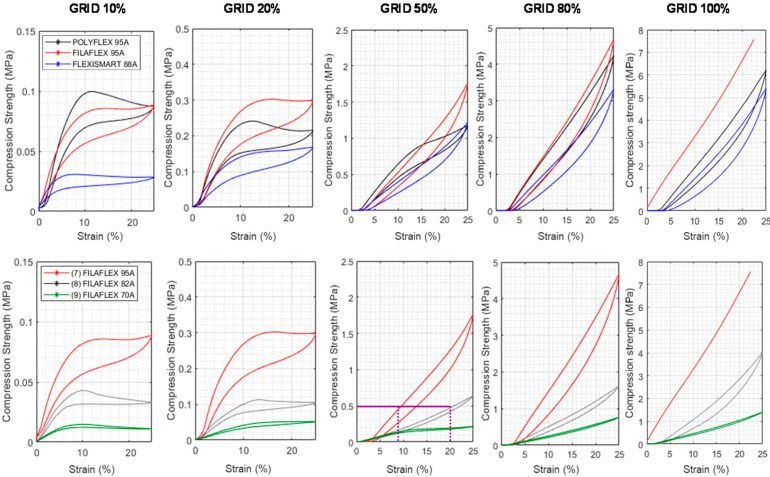
Compression strength measured for the different materials varying the infill density between 10 and 80%. The pattern type has been maintained (GRID).

**Figure 10 polymers-13-03551-f010:**
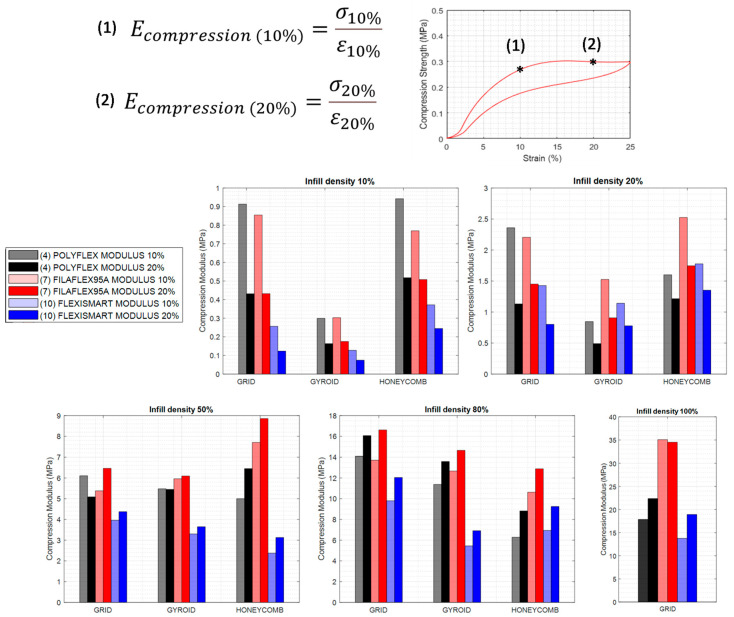
Compression modulus at 10% and 20% of deformation measured for specimens fabricated with Filaflex 95A, PolyFlex 95A and FlexiSmart. The measured specimens were fabricated with variable infill density and three different patterns (grid, gyroid and honeycomb).

**Figure 11 polymers-13-03551-f011:**
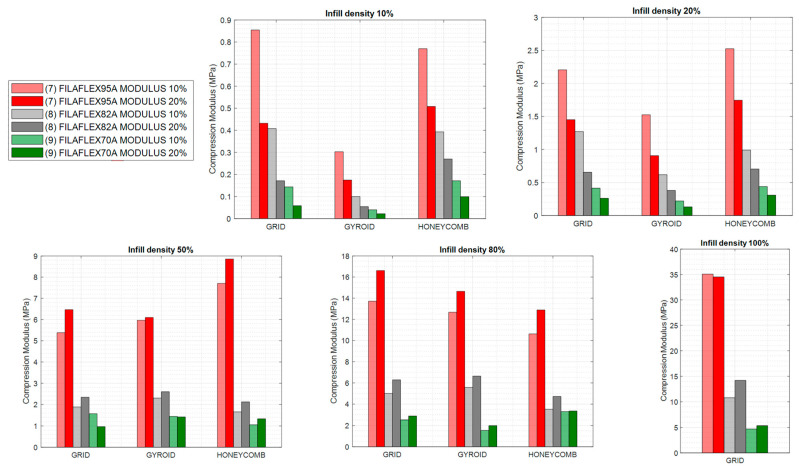
Compression modulus at 10% and 20% of deformation measured for specimens fabricated with Filaflex with different hardness (95 A, 82 A and 70 A). The measured specimens were fabricated with variable infill density and three different patterns (grid, gyroid and honeycomb).

**Figure 12 polymers-13-03551-f012:**
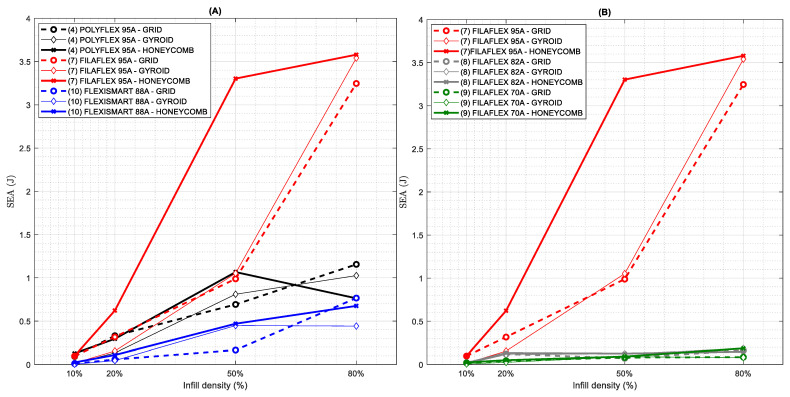
Specific Energy Absorption at 4th cycle as a function of the infill density. (**A**) TPUs with similar Shore harness: Filaflex 95 A, Polyflex 95 A and FlexiSmart 88 A, (**B**) Filaflex with different hardness: 95 A, 82 A and 70 A.

**Figure 13 polymers-13-03551-f013:**
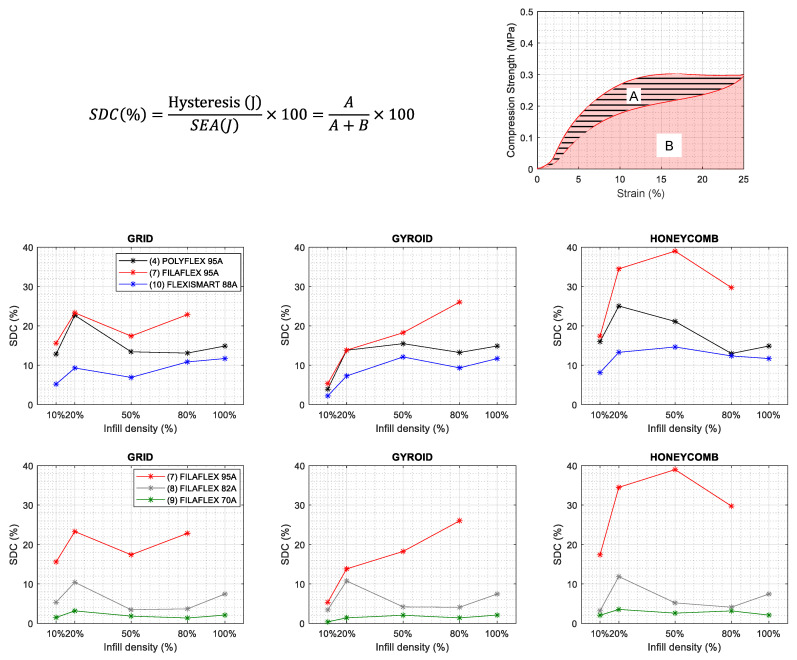
Specific Damping Capacity calculated from compression curves (4th cycle) as a function of the infill density. (A) TPUs with similar shore harness: Filaflex 95 A, Polyflex 95 A and FlexiSmart 88 A, (B) Filaflex with different shore: 95 A, 82 A and 70 A.

**Table 1 polymers-13-03551-t001:** Monomer composition, hard segment content and commercial name of the different TPU.

Monomer Composition	Hard Segment (%)	Shore	Commercial Name/Brand
PBA, MDI, BD	52.4	95 A	(1) TPU-Ultimaker (95 A)
	57.8	98 A	(2) Flexfill (98 A)
	42.0	93 A	(3) Smartfil Flex (93 A)
	44.6	95 A	(4) PolyFlex (95 A)
PTMG, MDI, BD	49.8	87 A	(5) eFLEX (87 A)
	85.1	83 D	(6) Innovatefil TPU Hardness + (83 D)
	54.2	95 A	(7) Filaflex (95 A)
	47.5	82 A	(8) Filaflex (82 A)
	31.4	70 A	(9) Filaflex (70 A)
PEA, MDI, BD	41.4	88 A	(10) Flexi Smart (88 A)

**Table 2 polymers-13-03551-t002:** Thermal properties of the TPUs: glass transition temperature (Tg), the fusion temperature (T_fusion_), temperature of 5% weight loss (T5%), temperature of maximum rate of decomposition of the first step (T_d1_).

Monomer Composition	Material	T_g_ (°C) (Initial to End)	T_fusion_ (°C) Hard Segment	T5% (°C)	T_d1_ (°C)
PBA, MDI, BD	(1) TPU-Ultimaker (95 A)	−41 (−54 to −34)	220 (10 to 238)	308.4	341.7
	(2) Flexfill (98 A)	−7 (−18 to +9)	174 (24 to 203)	304.6	342.0
	(3) Smartfil Flex (93 A)	−38 (−49 to −27)	198 (27 to 226)	302.0	350.8
	(4) PolyFlex (95 A)	−22 (−32 to −8)	160 (27 to 191)	310.9	346.0
PTMG, MDI, BD	(5) eFLEX (87 A)	−43 (−59 to −31)	165 (32 to 194)	303.1	321.3
	(6) Innovatefil TPU Hardness + (83 D)	+52 (+40 to +76)	207 (138 to 230)	301.0	334.1
	(7) Filaflex (95 A)	−37 (−57 to −19)	174 (27 to 218)	302.3	341.8
	(8) Filaflex (82 A)	−45 (−57 to −33)	163 (33 to 184)	299.0	326.8
	(9) Filaflex (70 A)	−60 (−69 to −51)	64 (−14 to 180)	289.7	297.9
PEA, MDI, BD	(10) Flexi Smart (88 A)	−27 (−35 to −19)	164 (28 to 191)	288.9	317.4

**Table 3 polymers-13-03551-t003:** Mechanical properties of the filaments of the different TPUs: Modulus (MPa), Tensile strength (MPa) and Elongation at break (%).

Material	Modulus (MPa)	Tensile Strength (MPa)	Elongation at Break (%)
(1) TPU-Ultimaker (95 A)	39 ± 2	44 ± 2	235 ± 14
(2) Flexfill (98 A)	137 ± 10	34.1 ± 1.0	405 ± 14
(3) Smartfil Flex (93 A)	46.7 ± 1.0	45.4 ± 1.0	490 ± 19
(4) PolyFlex (95 A)	28.6 ± 1.0	34.1 ± 1.4	640 ± 60
(5) eFLEX (87 A)	17.4 ± 0.4	30.4 ± 1.1	730 ± 20
(6) Innovatefil TPU Hardness + (83 D)	730 ± 50	51.1 ± 0.6	26 ± 7
(7) Filaflex (95 A)	31.4 ± 0.7	33 ± 3	610 ± 30
(8) Filaflex (82 A)	13.2 ± 0.3	33.6 ± 1.1	820 ± 20
(9) Filaflex (70 A)	5.5 ± 0.3	21 ± 2	1200 ± 160
(10) Flexi Smart (88 A)	22.5 ± 0.6	21.8 ± 0.8	1050 ± 50

**Table 4 polymers-13-03551-t004:** Main printing parameters selected for the fabrication of the 3D printed specimens.

Parameters	Polyflex95A	Filaflex95A	Filaflex82A	Filaflex70A	FlexiSmart88A
Nozzle diameter (mm)	0.4	0.4	0.4	0.4	0.4
Layer height (mm)	0.2	0.2	0.2	0.2	0.2
Extrusion temperature (°C)	225	225	230	235	225
Heat bed temperature	60	40	40	40	40
Extrusion speed (mm·s^−1^)	10–20	10–20	8–10	5–20	5–20
Fan speed (%)	-	-	-	100	-

## Data Availability

Not applicable.

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
