# Peer review of "3D Printing of Thermoplastic Elastomers: Role of the Chemical Composition and Printing Parameters in the Production of Parts with Controlled Energy Absorption and Damping Capacity"

_polymers, 2021, doi:10.3390/polym13203551_

Round 1

Reviewer 1 Report

The authors have used different commercial TPU filaments to study the influence of their chemical composition and geometry on mechanical properties, thermogravimetry analysis and energy absorption properties. TPU is most widely used polymer for the fabrication of scaffolds for biomedical applications. The current study will be very helpful for the researchers to find all the necessary information about TPU filaments in this article. Hence, I recommend this article for publication after minor revision. My comments are given below:

  1. Mention the full form of TPU in the abstract.
  2. Authors need to cite a few articles in the introduction about utilization of TPU filaments for different applications such as biomedical, aerospace, civil engineering etc. This will help readers to understand the possible use of TPU filaments in different fields.

Author Response

Comments and Suggestions for Authors

The authors have used different commercial TPU filaments to study the influence of their chemical composition and geometry on mechanical properties, thermogravimetry analysis and energy absorption properties. TPU is most widely used polymer for the fabrication of scaffolds for biomedical applications. The current study will be very helpful for the researchers to find all the necessary information about TPU filaments in this article. Hence, I recommend this article for publication after minor revision. My comments are given below:

  1. Mention the full form of TPU in the abstract.

According to the referee we have now included the full form of TPU, thermoplastic polyurethanes, in the abstract

Line [19]: … Commercial thermoplastic polyurethanes (TPU) filaments …

  1. Authors need to cite a few articles in the introduction about utilization of TPU filaments for different applications such as biomedical, aerospace, civil engineering etc. This will help readers to understand the possible use of TPU filaments in different fields.

In this new version of the manuscript, examples of applications of TPU have been added for biomedical applications, sports and engineering following reviewer considerations.

Line [74]: TPUs for applications such as biomedical [29-32], sports equipment [33] or engineering applications [21,34,35].

Reviewer 2 Report

The paper titled "3D Printing of Thermoplastic Elastomers: Role of the Chemical Composition and Printing Parameters in the Production of Parts with Controlled Energy Absorption and Damping Capacity", by Marina León-Calero et al. is, in my opinion, not sufficiently sound for publication in the journal. The following suggestions should be considered:

  1. The Abstract section, you’d better present the problems to be solved and the corresponding results of your experiments. What’s more, the keywords are too many and please modify.
  2. In Introduction section, you’d better introduce detailedly the advantages of Additive manufacturing and Thermoplastic Elastomers. You should mention the strategies others have attempted or what successes have been achieved about them.
  3. I would suggest the authors revealing the benefits of FDM compared to alternative approaches such as SLS to prepare thermoplastic elastomers, e.g. Physics Procedia, 2016, 83: 971-980, Bioactive Materials, 2021, 6:490-502.
  4. In Materials and Methods section, you should describe the process of 3D printing in detail with an example. And show that which experiments can explain controlled energy absorption and damping capacity.
  5. The hard segment content of the different TPU are shown in Table 1 and please explain the difference and influence on the experiments.
  6. Please report the temperature at which the mass loss rate is maximum and discuss the results.
  7. Please draw a conclusion of best 3D printing parameters such as infill densityand infill pattern and their influence on the properties of specimens, such as specific energy absorption and damping capacity.

Author Response

Comments and Suggestions for Authors

The paper titled "3D Printing of Thermoplastic Elastomers: Role of the Chemical Composition and Printing Parameters in the Production of Parts with Controlled Energy Absorption and Damping Capacity", by Marina León-Calero et al. is, in my opinion, not sufficiently sound for publication in the journal. The following suggestions should be considered:

  1. The Abstract section, you’d better present the problems to be solved and the corresponding results of your experiments. What’s more, the keywords are too many and please modify.

In order to better present the problems to be solved in the paper and the corresponding results of the experiments, the Abstract has been improved. In this version we clarified that there are only few examples in which the mechanical properties of the 3D printed parts have been correlated with the fabrication parameters. Moreover, there is no precedent for a thorough chemical characterization of commercially available thermoplastic elastomers filaments. We took advantage of this know how in the chemical composition to design the 3D printed parts with the best absorption and damping capacities.

Regarding the problem to be solved we added this new sentence: Previous investigations have exclusively focus on the optimization of the printing parameters of commercial TPE filaments and the structures to analyse the mechanical properties of the 3D printed parts. In the present paper, the chemical, thermal and mechanical properties for a wide range of commercial thermoplastic polyurethanes (TPU) filaments were investigated.

Furthermore, for the results of the experiments, the best condition has been included in the Abstract: “As a result, optimal SEA and SDC performances were obtained for honeycomb pattern at a 50% of infill density”

In agreement with the referee we have now reduced the number of keywords: thermoplastic elastomers, fused deposition modeling, polymer characterization, mechanical properties, energy absorption, damping capacity.

  1. In Introduction section, you’d better introduce detailedly the advantages of Additive manufacturing and Thermoplastic Elastomers. You should mention the strategies others have attempted or what successes have been achieved about them.

The advantages of additive manufacturing, included in the first paragraph, have been more detailed to further clarify this aspect: Additive manufacturing (AM), in contrast to convectional technologies such as injection or extrusion, involves layer-by-layer construction, so that upon the subsequent addition of multiple layers a 3D structure is obtained. In comparison to the above mentioned techniques, AM enables de fabrication of highly complex structures defined by a CAD design, in a shorter period of time and, with a clear material waste reduction.

Regarding the second concern of the referee we would like to mention that the advantages of thermoplastic elastomers were already included in Line 55. The text is the following: “Thermoplastic elastomers are of high interest due to their ability to stretch to moderate elongations and return to its near original shape, energy absorption capacity [10-13], energy efficiency [10, 11], cushion factor [14, 15] or damping capacity [14]”.

  1. I would suggest the authors revealing the benefits of FDM compared to alternative approaches such as SLS to prepare thermoplastic elastomers, e.g. Physics Procedia, 2016, 83: 971-980, Bioactive Materials, 2021, 6:490-502.

As indicated by the referee, in this new version we included a new paragraph in which the benefits of FDM compared to SLS have been indicated. More precisely, in Line [59] up to line 62: It is worth mentioning that thermoplastic elastomer parts can be also manufactured by Selective Laser Sintering (SLS) technology. However, SLS presents some disadvantages with respect to FDM such as higher cost, low recyclability, more complex processability or design limitations (such as in the case of the design of closed cavities) [16].

The reference in SLS has been equally included as suggested by the referee. However, we did not find any relation with reference two (Bioactive Materials), so we decided not to include this one.

  1. In Materials and Methods section, you should describe the process of 3D printing in detail with an example. And show that which experiments can explain controlled energy absorption and damping capacity.

According to reviewer FDM printing process have been included in 2.8: “Fused deposition modelling (FDM) is a 3d printing process in which a thermoplastic passes through a heating element that partially melts the material. This semi-molten material is then dropped through a nozzle which can move in the XY plane, onto a platform. The deposition takes place according to a path which is generated by the slicing software. After completing one layer of deposition, the part moves vertically in the Z plane and the process repeats until the part is finished”.

We also clarified that the final optimized parameters for the printing process are summarized in Table 4

We apologize for the misunderstanding in the explanation of the estimation of the absorption energy and damping capacity. Regarding these experiments, the values of the energy absorption and damping capacity were obtained from the calculations indicated in Figure 5 (supporting information) for the case of Specific Energy Absorption and in Figure 13 for the case of Specific Damping Capacity. These values were calculated from the stress-strain curves obtained in the compression tests. An explanation about the compression tests is provided in 2.7. For clarity purposes we included the following sentence: “These mechanical tests allow to calculate mechanical properties such as the compression modulus at different elongation conditions, hysteresis, energy absorption capacity or specific damping capacity for each material, pattern and infill densities”

  1. The hard segment content of the different TPU are shown in Table 1 and please explain the difference and influence on the experiments.

According to the referee suggestion we attempt in this new version to highlight the parts in which we establish a direct relation between the HS content and the final properties.

As mentioned by the referee in Table 1 we provide the values of hard segment obtained for the different materials investigated. The calculation of these values was made by using the 1H NMR spectra that allowed us to determine the chemical composition of the TPU and therefore the amount of soft and hard parts. In line 253 we already explained how this values were obtained: “TPUs are segmented polymers where the macroglycol chains constitute the so-called soft segments (SS) whereas the segments produced by the reaction of the diisocyanate MDI with the chain extender BD constitute the so-called hard segments (HS)”

In addition to the calculation of the HS content we provided along the paper explanations about the relation between the hard segment content and the mechanical properties of the 3D printed parts.

See, for instance:

Line 438: “However, these differences in the plateau height can be related to the hard segment content which is significantly higher for Filaflex (54.2%) than for Polyflex (44.6%).”

Line 589: “This is a relevant result since in addition to the Shore hardness, the chemical structure of the material together with the hard segment content appears to be determinant in the final properties of the printed part.”

Considering the referee suggestion, detailed explanation have been included related to the influence on the experiments.

A preliminary information about the relation between HS and mechanical properties has been included in Line [268]: “Notice that, independently of the shore grade, the %HS have an interesting impact on the mechanical properties of the 3D printed parts providing, for example, higher strength for a higher % of the hard segment”.

In addition, extra explanations for the mechanical result with respect to the 3d printing parameters have been included.

Line 447 “Furthermore, FlexiSmart presents lower strength at compression than Polyflex which can be related to the lower %HS content of FlexiSmart (41.6%)”.

Line 511: “It is interesting to note that, although Filaflex95A and Polyflex95A presents the same Shore hardness, the compression modulus shows slight differences which might be associated to the higher %HS of Filaflex95A (54.2%) respect to Polyflex95A (44.6%)”

Line 557: “...depends on the type of material (and its related %HS content) and the infill pattern.”

  1. Please report the temperature at which the mass loss rate is maximum and discuss the results.

Regarding this comment, we have to mention that the values requested by the referee are already included in Table 2. More precisely, this table has a column Td1 referring to the “temperature of maximum rate of decomposition”.

Concerning the explanation of the decomposition process we included previous references in which this aspect has been investigated. “Above this range of temperatures thermal decomposition takes place in two distinct steps. As it has been described in literature, polyurethanes usually decompose in two main steps with the first step due to the decomposition of the urethane groups in the main chain and the second to the decomposition of the rest of the material (i.e. C-C and C-O bonds)”.

Nevertheless a new paragraph has been included in which we compare the results obtained for the different TPUs investigated. While this occurs for all the TPUs some differences were observed depending on the chemical composition. More precisely, the TPUs formed using PBA as macroglycol presented the higher degradation temperatures in the range of 340-350°C. Those pre-pared from PTMG presented Td1 in the range of 320-340°C (for the range of hardness between 87A and 95A) and finally the TPU prepared from PEA presented the lower value at around 317°C. It is also interesting to note that for the same chemical composition a reduction in the HS content has associated a decrease in the Td1. This can be clearly evidenced in the case of the different Filaflex. Filaflex 85A has a Td1 of ~341°C, which is reduced to ~327°C for the case of Filaflex 82A and finally to 298°C for Filaflex 70A.

  1. Please draw a conclusion of best 3D printing parameters such as infill density and infill pattern and their influence on the properties of specimens, such as specific energy absorption and damping capacity.

As suggested by the referee, a paragraph has been included in the conclusions part providing the relation between the type of material (%HS content) and 3d printing parameters (infill pattern and infill density) with respect to the specify energy absorption and damping capacity:

Line 630: “Regarding the type of material, and specially its %HS content, it has been observed that for higher %HS content (Filaflex95A), the energy absorption and damping capacity are in-creased. In addition, for higher infill density, the absorption energy is increased; however, the damping capacity improvement de-pends on the infill pattern and does not show a direct relation to the infill density. Lastly, honeycomb pattern shows the best results for SEA and SDC in the range of 20-80% of infill density”.

Round 2

Reviewer 2 Report

The revised manuscript has addressed my previous concerns.

Author Response

Dear editor,

we have revised the language and style in order to remove all grammatical mistakes.

We provide two versions:

 - With marks: in order to follow the changes made

  • Without marks: the final version clean of comments and marks

We hope you find this version appropriate to be published in Polymers

Best regards

Juan